# Hierarchical marker genes selection in scRNA-seq analysis

**Yutong Sun[1], Peng Qiu[2]***

**1** School of Electrical and Computer Engineering, Georgia Institute of Technology, Atlanta, Georgia, United States of America, **2** Department of Biomedical Engineering, Georgia Institute of Technology and Emory University, Atlanta, Georgia, United States of America

* peng.qiu@bme.gatech.edu

## Abstract

When analyzing scRNA-seq data containing heterogeneous cell populations, an important task is to select informative marker genes to distinguish various cell clusters and annotate the clusters with biologically meaningful cell types. In existing analysis methods and pipelines, marker genes are typically identified using a one-vs-all strategy, examining differential expression between one cell cluster versus the combination of all other cell clusters. However, this strategy applied to cell clusters belonging to closely related cell types often generates overlapping marker genes, which capture the common signature of closely related cell clusters but provide limited information for distinguishing them. To address the limitations of the one-vs-all strategy, we propose a hierarchical marker gene selection strategy that groups similar cell clusters and selects marker genes in a hierarchical manner. This strategy is able to improve the accuracy and interpretability of cell type identification in single-cell RNA-seq data.

**Data Availability Statement:** Code and data are available at \url{https://github.com/syt960909/Hierarchical-Maker-Genes-Selection-for-scRNA-seq-Data}.

**Funding:** This publication is part of the Gut Cell Atlas Crohn's Disease Consortium funded by The Leona M. and Harry B. Helmsley Charitable Trust

## Author summary

In the analysis and interpretation of scRNA-seq data, one important step is to identify marker genes to annotate cell clusters with the biologically meaningful names. Existing marker gene selection methods typically perform differential expression between one cell cluster versus all other clusters combined. Ideally, marker genes for one cell cluster should be highly expressed in the cell cluster and lowly expressed in the other cell clusters. However, when there exist cell clusters that correspond to closely related cell types, the one-vs-all approach often introduces overlapping marker genes that represent the commonality among the closely related cell types but provide limited information to interpret their differences. Here we organize cell clusters in a hierarchical manner, and define marker genes at all levels of the hierarchy. Our approach provide marker genes not only for individual clusters but also for lineages defined by closely related clusters. The proposed hierarchical marker genes are able to better separate cell types and better facilitate cell type annotation across datasets in those biological contexts.

and is supported by a grant from Helmsley to Georgia Institute of Technology (www.helmsleytrust.org/gut-cell-atlas/). This work was also supported by the National Science Foundation (CCF2007029). The funders had no role in study design, data collection and analysis, the decision to publish, or the preparation of the manuscript.

**Competing interests:** The authors have declared that no competing interests exist.

## Introduction

As scRNA-seq technologies continue to advance, analyzing gene expression patterns at the single-cell level has become an increasingly popular approach to understand cellular heterogeneity and differentiation [1, 2]. Identification of marker genes is a crucial component of the analysis, as the marker genes allow us to distinguish various cell clusters and identify their cell types and states based on their unique gene expression signatures [3].

Selection of marker genes is a non-trivial task. In the literature, existing marker gene selection approaches can be organized into two categories. One category adopts a one-vs-all strategy [4–6], while the other uses a hierarchical strategy [7]. The one-vs-all strategy is more commonly used to identify marker genes in scRNA-seq data. Methods adopting this strategy aim to identify marker genes that exhibit differential expression between one cell cluster and the combination of other cell clusters. For example, Seurat [4] identifies differentially expressed genes in each cell cluster compared to all other cells in the dataset using a Wilcoxon rank sum test. Genes with the highest differential expression are selected as marker genes for each cell cluster. Monocle [5] uses logistic regression to identify genes that are differentially expressed between each cell cluster and all other cells in the dataset. Genes with the highest probability of being expressed in a specific cell cluster are selected as marker genes. There are several ranking-based one-vs-all marker gene selection methods. SingleR [6] compares gene expression profiles in scRNA-seq datasets to reference bulk transcriptomic data of sorted cell clusters to identify marker genes that are differentially expressed between the target cell cluster and all other cell clusters. SingleR employs a ranking-based method that takes into account the degree of differential expression and the prevalence of the marker gene in the target cell cluster. The top-ranked genes for each cell cluster are selected as the marker genes for that cell cluster. COMET [3] is a ranking-based brute force approach that selects sets of up to four markers with the best predictive power to separate one cell cluster. RankCorr [8] applies a rank sum transformation that provides a non-parametric way of considering the counts eliminates the need to normalize the data, and selects an informative number of markers for each cluster in a one-vs-all fashion. Several existing methods apply consensus optimization to select marker genes. SC3 [9] proposes an unsupervised consensus clustering approach by combine binary classification based on mean cluster-expression values and Wilcoxon signed-rank test to compute p-values. Genes with high areas under ROC curve and low p-value are selected as marker genes. Both SCMarker [10] and scTIM [11] use consensus optimization strategy to identify marker genes. SCMarker uses a mixture distribution model to select genes that are individually discriminative across underlying cell clusters and are either co-expressed or mutually exclusively expressed with other genes. scTIM integrates 'gene specificity', 'cell relation network', and 'gene redundancy' into a multi-objective optimization problem. SMaSH filters and ranks genes according to an ensemble learning model or a deep neural network. These one-vs-all marker gene selection approaches do not take advantage of the hierarchical relationships that exist among cell clusters and the correlations in expression patterns among genes, which are crucial aspects in obtaining a comprehensive understanding of cell cluster identities and biological processes [7].

A relatively less popular strategy for marker gene selection is to incorporate the hierarchical structure of cell clusters, which not only identifies marker genes for individual cell clusters (leaves of a hierarchical tree), but also provides marker genes for subsets of closely related cell clusters (intermediate nodes of the hierarchy). This strategy has the potential to provide more interpretable markers, as genes that are differentially expressed across multiple subsets of cell clusters are more likely to be involved in key biological processes. One such algorithm is scGeneFit [7], which combined multi-variate projection and hierarchical ideas. Given a cell cluster

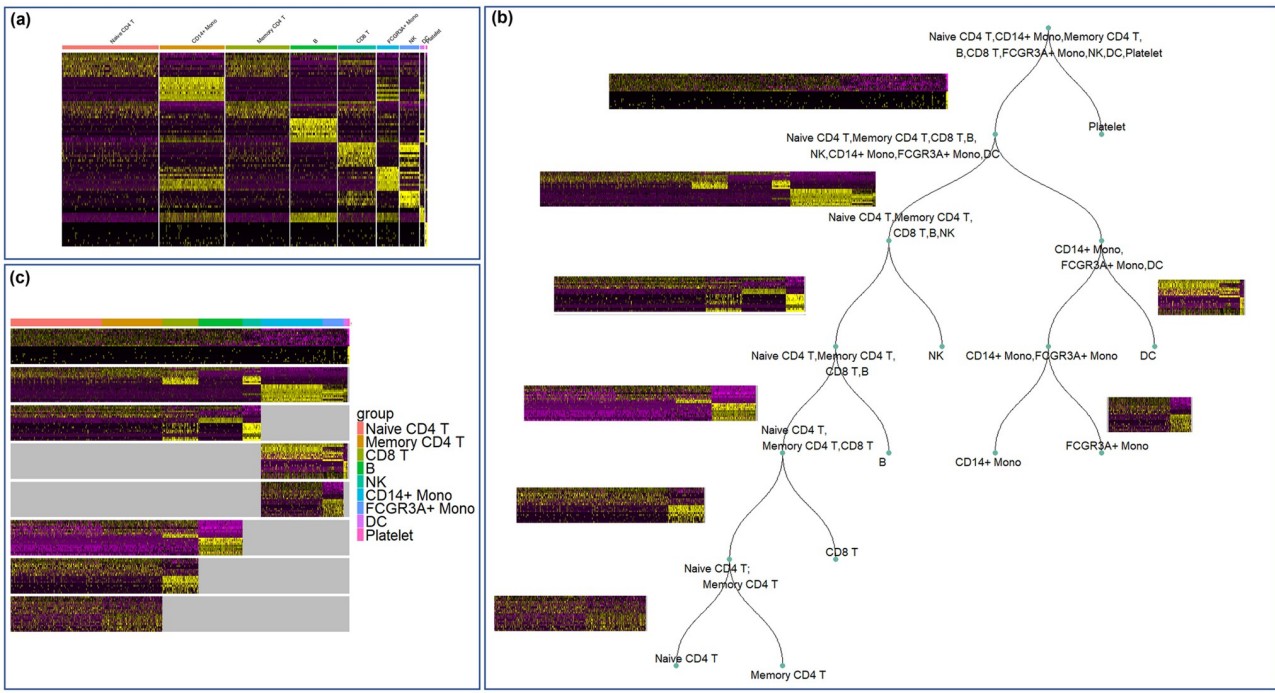

**Fig 1. Overview of hierarchical marker gene selection in PBMC3k data. (a)** Marker gene heatmap generated by the one-vs-all FindMarker approach in Seurat. **(b)** Our constructed hierarchical structure of cell clusters in the PBMC3k dataset. **(c)** Assembled heatmap that concatenates marker gene heatmaps for individual splits in the constructed cell cluster hierarchy.

hierarchy provided by expert users or hierarchical clustering algorithms, scGeneFit solves one projection problem for each split in the hierarchy, where each projection problem aims to find to the lowest-dimensional subspace to separate cells in different classes defined by the corresponding split in the hierarchy. In scGeneFit, defining the cell cluster hierarchy and finding marker genes are considered separately and consecutively, and hence, the definition of cell cluster hierarchy is not affected by the subsequent analysis of finding marker genes.

In this paper, we explore a hierarchical approach for finding marker genes, where the definition of cell cluster hierarchy and the identification of marker genes are jointly considered. Our approach is motivated by one drawback of the one-vs-all strategy, which often generates overlapping marker genes for closely related cell clusters, capturing their common signature but providing limited information to distinguish them. One example is shown in Fig 1a. When one-vs-all cell cluster marker genes are visualized using heatmaps, diagonal blocks of high expression confirm that the identified cell cluster marker genes are indeed highly expressed in the corresponding cell clusters, while off-diagonal blocks of high expression indicate that marker genes for one cell cluster may also be highly expressed in other cell clusters which is undesirable. Therefore, we propose to identify an optimal grouping of cell clusters that minimizes off-diagonal expression signal, so that the one-vs-all strategy identifies marker genes specific to each cell cluster group. Within each cell cluster group, the same analysis can be applied to identify optimal subgroups, so that the one-vs-all strategy produces marker genes highly specific to each subgroup. Iterating such off-diagonal minimization analysis induces a cell cluster hierarchy, as well as marker genes for the cell clusters or cell cluster groups at each split of the hierarchy. Using real scRNA-seq datasets, we compared our hierarchical approach with the one-vs-all marker selection approach in Seurat and the hierarchical approach in

scGeneFit, and demonstrated the advantage of our approach in terms of its interpretability and its performance in automated cell type mapping.

## Results

### Hierarchical marker genes selection framework

To motivate the proposed hierarchical marker gene selection framework, we used an example scRNA-seq dataset of PBMC, which contained data for 2638 cells grouped into 9 cell clusters that correspond to 9 cell types. The cell types included Naive CD4 T cells, CD14+ Monocytes, Memory CD4 T cells, B cells, CD8 T cells, FCGR3A+ Monocytes, NK cells, Dendritic cells and Platelets. Applying the one-vs-all find marker approach implemented in Seurat, the expression patterns of the identified markers were shown as the heatmap in Fig 1a, where each vertical section corresponds to one cell cluster labeled by its cell type name. In addition to the diagonal blocks of high expression that confirmed the expression of identified marker genes in their corresponding cell clusters, multiple off-diagonal blocks of high expression were observed. For example, the marker genes for the first cell cluster (Naive CD4 T cells) were also highly expressed in the third cell cluster (Memory CD4 T cells). This is reasonable because the two CD4 T cell subtypes are closely related. However, these marker genes may not provide sufficient information to interpret and separate these two clusters corresponding to the two closely related CD4 T cell subtypes.

We propose a scoring function defined as the average of diagonal expression minus the average of off-diagonal expression (details in Methods). This scoring function quantifies how much undesirable off-diagonal expression exists in the marker gene heatmap. We then combine two of the cell clusters, re-do the one-vs-all marker gene identification to re-generate the marker gene heatmap, and use the scoring functions to quantify how much off-diagonal expression exists after the two cell clusters are combined. We examine all possible pairs of cell clusters to find the best pair whose combination leads to the least off-diagonal expression in the marker gene heatmap. If combining this best cell cluster pair is able reduce off-diagonal expression compared to not combining them, we merge this pair of cell clusters, so that the number of cell clusters reduces by one. After that, we perform the same analysis to the resulting cell clusters, identify the best pair of cell clusters whose combination leads to least off-diagonal expression, and merge this pair if the off-diagonal expression is further reduced after merging. This process iterates until no merge is able to further reduce off-diagonal expression in the marker gene heatmap. This is essentially an agglomerative clustering process of the cell clusters, using the proposed scoring function as both distance metric and stopping criterion. In this example dataset, the agglomerative process stopped when the 9 original cell clusters were merged into two: one was the cell cluster corresponding to Platelets and the other was the remaining 8 cell clusters combined. As shown in Fig 1b, the first split of our cell clusters hierarchy had two branches, separating the Platelets and all other cell types. Performing one-vs-all marker finding for these two branches produced marker genes for Platelets and marker genes for other cell types combined, and the resulting heatmap is shown next to the first split in Fig 1b.

To further construct the cell cluster hierarchy, we focused on the 8 cell clusters belonging to the left branch of first split, as if we were analyzing a new dataset composed of these 8 cell clusters. We agglomeratively merged these 8 cell clusters, and used the scoring function to stop the agglomeration when the off-diagonal expression was minimized. In this example, the 8 cell clusters were agglomeratively merged into two groups, so that the second split of the cell cluster hierarchy was also a two-way split: one branch was the combination of the 3 myeloid cell clusters (Monocytes and Dendritic cells), and the other branch was the combination of the

remaining 5 lymphoid cell clusters (B cells, T cells and NK cells). Similarly, the branch containing the 3 myeloid cell clusters and the branch containing the 5 lymphoid cell clusters were examined separately to construct additional splits in the cell cluster hierarchy. This construction process iterated until all of the 9 original cell clusters were separated as leaf nodes in the cell cluster hierarchy. In summary, our cell cluster hierarchy is essentially a divisive hierarchical clustering process, where each split is determined by an agglormerative process to minimize undesirable off-diagonal expression and hence maximize specificity of the identified marker genes.

The marker gene heatmaps for individual splits in the cell cluster hierarchy can be concatenated and assembled into Fig 1c, where each horizontal section corresponds to an individual split named by the corresponding cell types within the split. Since majority of the splits only considered a subset of the cell clusters, for a particular horizontal section corresponding to one split, expression data for cell clusters not considered in the split were zeroed out and shown as white areas in the assembled heatmap. This visualization provides a compact view of marker genes defined by our hierarchical approach.

## Data collections for evaluation

To evaluate the proposed hierarchical marker gene selection approach, we applied it to three peripheral blood mononuclear cells (PBMC) datasets, namely PBMC3k [12], PBMC control, and PBMC stimuated [13]. PBMC3k is the dataset used as the illustrative example in Fig 1. The PBMC control dataset contained 6573 cells, and the PBMC stimulated dataset contained 7263 cells. Cells in both of these two datasets were grouped into 13 cell clusters. In addition to the PBMC datasets, we also included analysis based on a human pancreas dataset published by Xin [14, 15], which contained 1492 cells grouped into 4 cell clusters. The purpose is to provide an example dataset where the proposed hierarchical marker selection approach degenerated to the flat one-vs-all approach, because the cell clusters were sufficiently distinct and the flat one-vs-all approach did not produce much undesirable off-diagonal expression signal in the marker expression heatmap.

## Hierarchical marker genes capture more cell type differences

We applied the proposed hierarchical marker gene selection to the three PBMC datasets (PBMC3k, PBMC control, PBMC stim). The hierarchical marker genes for PBMC3k data are visualized in Fig 1b. The hierarchical marker genes for the other two PBMC datasets are shown in Figs 2b and 3b. In addition, we applied the one-vs-all marker finding approach in Seurat [16], and both the flat and hierarchical versions of scGeneFit [7] to the three datasets. We compared the selected genes, as well as directly using all genes or the highly variable genes, in terms of their ability to separate the cell types annotated in these datasets. More specifically, each dataset was split into a training set and a testing set with a 7:3 ratio. K-Nearest Neighbor classifiers were trained based on genes selected by various approaches, and the classification accuracies were evaluated using the testing set.

For the PBMC3k dataset, the classification accuracies are shown in Fig 4a. The first 9 sets of color bars show cell-type-wise classification accuracies. "Average" denotes the average value of cell-type-wise classification accuracies across all cell types. We observed that the cell-type-wise classification performance varied across cell types. As baseline references, classification based on all genes or the top 1406 highly variable genes (the number of highly variable genes were determined automatically by Scanpy with its default parameters) achieved accuracy between 39% and 98% for various cell types, with an average accuracy of around 80%. The one-vs-all marker genes from Seurat improved classification accuracy by around 4%, even though the

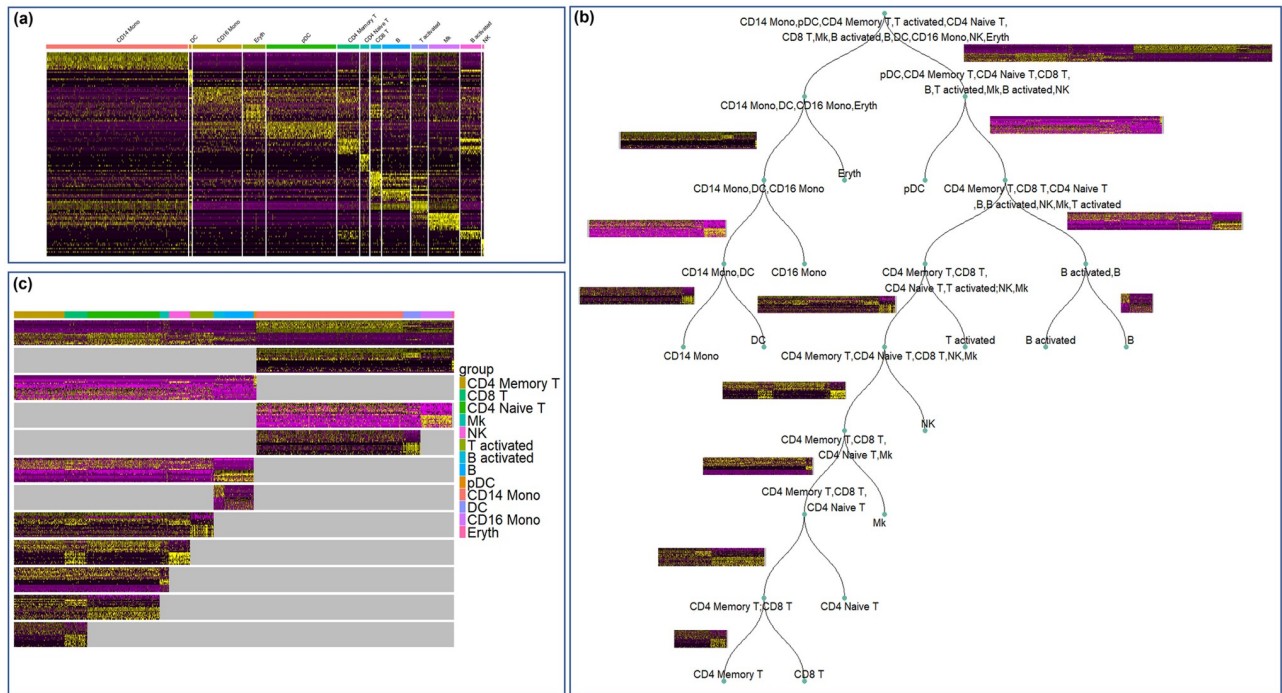

**Fig 2. Hierarchical marker gene selection in PBMC control dataset. (a)** Marker gene heatmap generated by the one-vs-all FindMarker approach in Seurat. **(b)** Constructed hierarchy of cell clusters in PBMC control dataset. **(c)** Assembled heatmap that summarizes all marker genes for various splits in the cell cluster hierarchy.

number of the one-vs-all marker genes was 118, far fewer than the highly variable genes. The marker genes generated by the flat and hierarchical versions of scGeneFit did not outperform the baselines. Finally, with the same number of marker genes compared to the one-vs-all approach in Seurat, the proposed hierarchical marker genes achieved the highest classification performance, which is 10.5% improvement over marker genes found by one-vs-all approach in Seurat.

The marker genes from the PBMC3k dataset were also evaluated based on UMAP visualizations in Fig 5a colored by cell types. UMAP visualization based on all genes is shown in the first column of Fig 5a, where major lineages were well separated but closely related cell types were co-located. More specifically, T cell subtypes formed one island, Monocytes formed one island, and B cells formed its own island. In the second and third columns of Fig 5a, UMAP based on high variable genes and one-vs-all marker genes found by Seurat produced tighter clusters and better separation among the major lineages. Although closely related cell types were still co-located, CD8 T cells were better separated from Naive CD4 T and Memory CD4 T. In the fourth and fifth columns, UMAP based on flat and hierarchical versions of scGeneFit showed poor cell type separation, consistent with the evaluation based on classification performance in Fig 4a. The last column of Fig 5a showed UMAP based on the proposed hierarchical marker genes. This UMAP was drawn based on the assembled data behind the heatmap in Fig 1c. In the last column of Fig 5a, we observed that many closely related cell types formed their own clusters. For examples, CD8 T cells, CD4 T cells and NK cells formed three isolated clusters, Monocytes and Dendritic cells formed two isolated clusters, while these cell types were co-located in UMAPs based on other marker gene selection methods. Although we still observed co-localization of Memory CD4 T and Naive CD4 T and co-localization of the two

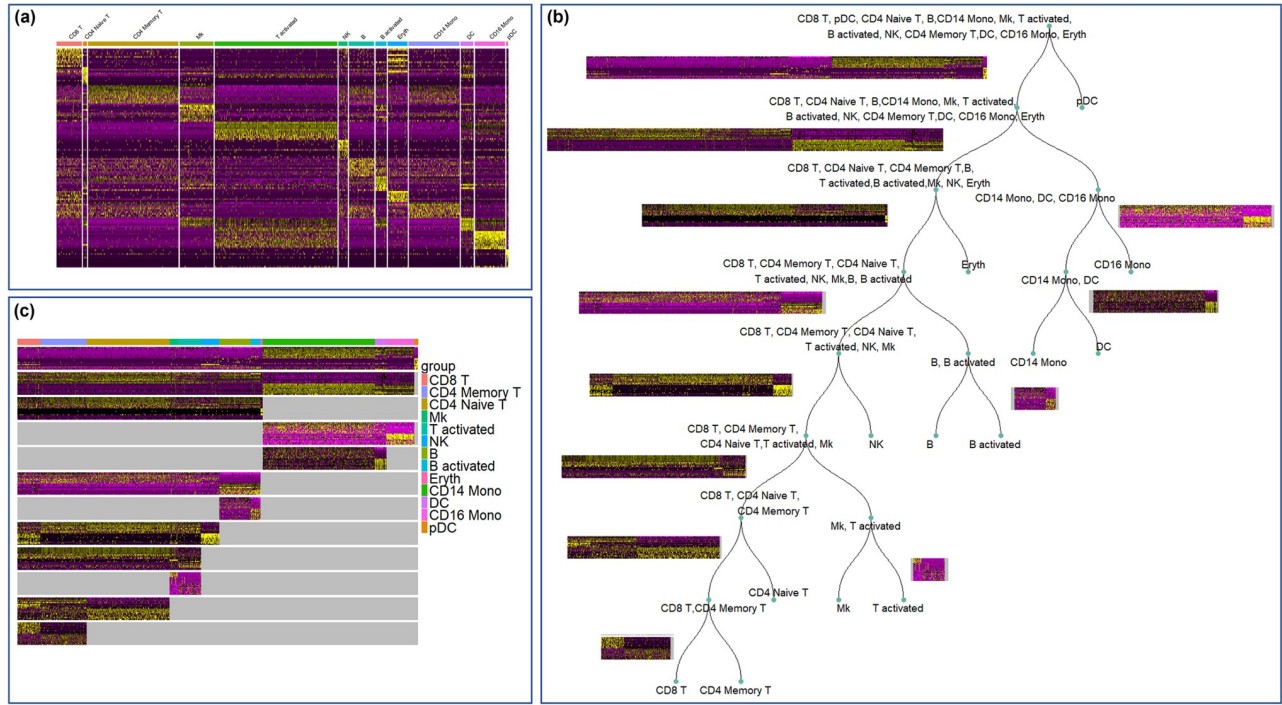

**Fig 3. Hierarchical marker gene selection in PBMC stim dataset. (a)** Marker gene heatmap generated by the one-vs-all FindMarker approach in Seurat. **(b)** Constructed hierarchy of cell clusters in PBMC stim dataset.**(c)** Assembled heatmap that summarizes all marker genes for various splits in the cell cluster hierarchy.

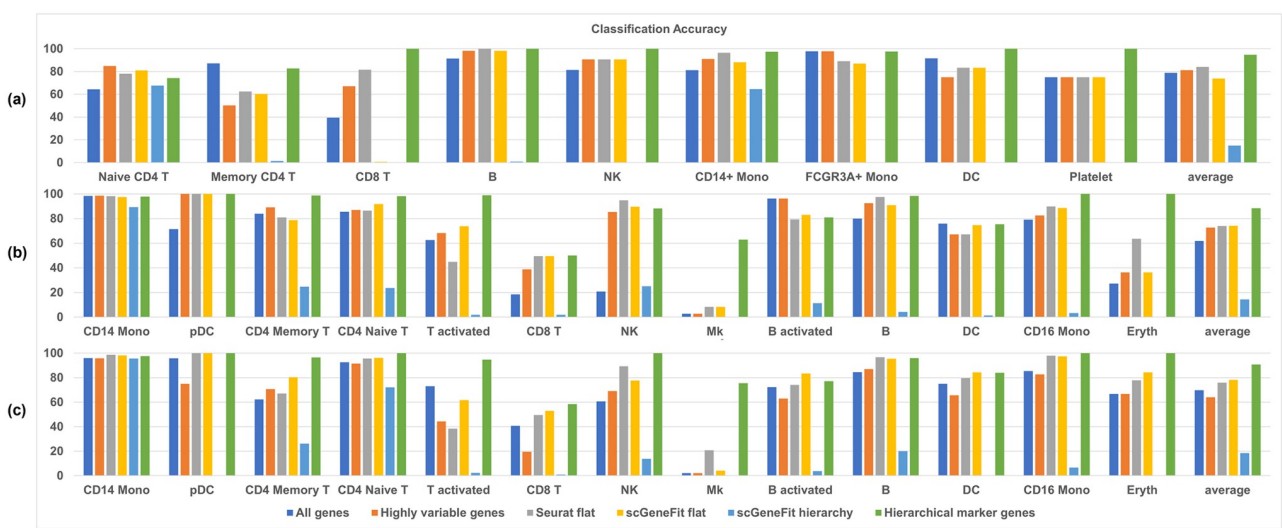

**Fig 4. Comparison of hierarchical marker genes with two baselines and three existing marker genes selection methods. Baselines are either all genes or highly variable genes. The three existing approaches are the flat one-vs-all FindMarker in Seurat, the flat version of scGeneFit, and the hierarchical version of scGeneFit. For each evaluation datasets, we trained a K-Nearest Neighbor classifier on 70% of the cells, and tested classification accuracy on the remaining 30% cells. (a)** Classification accuracies for the PBMC3k dataset; **(b)** Classification accuracies for the PBMC control dataset; **(c)** Classification accuracies for the PBMC stim dataset.

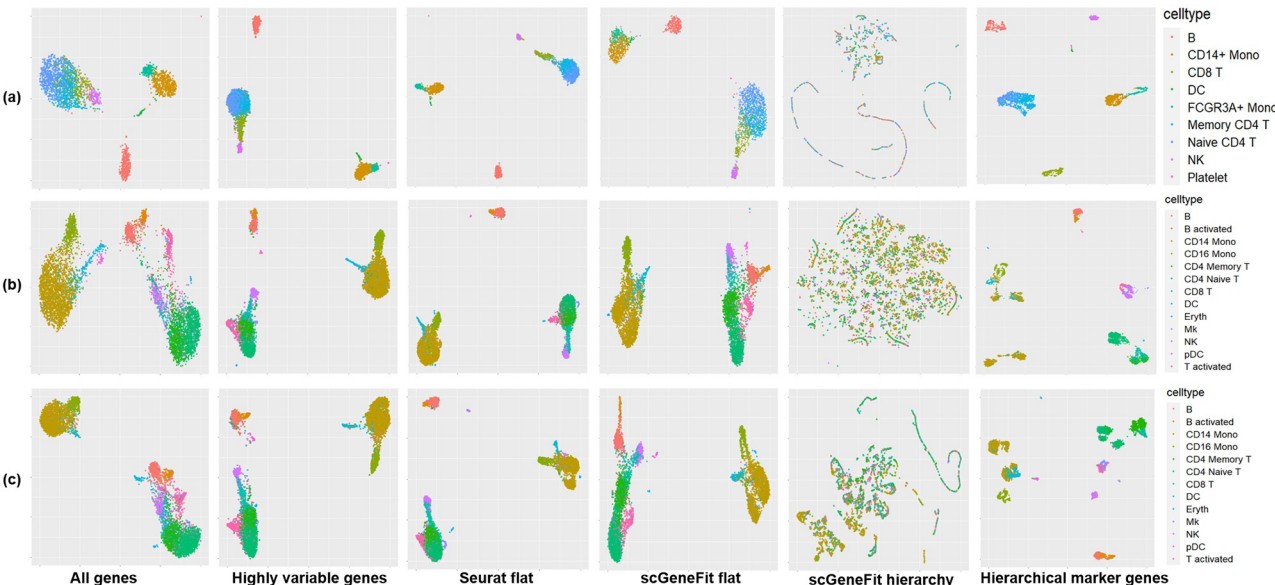

**Fig 5. UMAP visualization of hierarchical marker genes, two baselines and three existing marker genes selection methods, applied to three datasets. (a)** UMAP visualizations of PBMC3k dataset colored by cell types; **(b)** UMAP visualizations of PBMC control dataset; **(c)** UMAP visualizations of PBMC stim dataset.

Monocyte subtypes, UMAP based on our hierarchical marker genes showed significantly better cell type separation compared to UMAPs based on other gene selection algorithms, which was consistent to the comparison based on classification accuracies.

For the PBMC control dataset, evaluation based on classification accuracy is shown in Fig 4b. We noticed that classification based on the proposed hierarchical marker genes achieved the highest average accuracy of 88%, while classification based on the one-vs-all marker genes from Seurat achieved an average accuracy of 74%. This difference is mainly contributed by the fact that the proposed hierarchical marker genes achieved much higher classification accuracies than the one-vs-all marker genes for several cell types, including T activated, CD16 Mono, Eryth and Mk. The marker genes in the PBMC control dataset were also visualized using UMAP shown in Fig 5b. Once again, based on the one-vs-all marker genes from Seurat (third column), UMAP showed islands that separated major lineages, while closely related cell types were co-located. In contrast, in the last column of Fig 5b, UMAP based on the hierarchical marker genes showed more islands that separated more cell types. Similar result was also observed in the PBMC stim dataset, as shown in Figs 4c and 5c.

## Hierarchical marker genes improve automated cell type mapping

We further compared various marker gene selection approaches in the context of cell type annotation, using the PBMC control dataset and the PBMC stim dataset. We first considered the PBMC control dataset as reference, applied various marker gene selection approaches to the reference dataset, and then used the reference dataset to train K-Nearest-Neighbors classifiers based on marker genes selected by those approaches. After that, we applied the classifiers to predict the cell type labels of cells in the PBMC stim dataset as query data. The resulting prediction accuracies are shown in Fig 6a. We noticed that the prediction accuracies varied across cell types. In average, the one-vs-all marker genes from Seurat achieved an average accuracy of 72.3%, while the hierarchical marker genes achieved an average prediction accuracy of 88.1%.

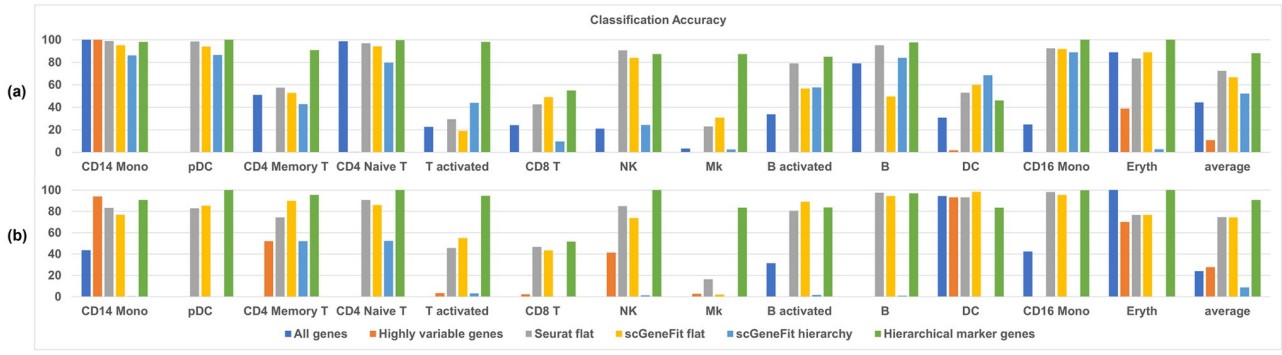

**Fig 6. Compare hierarchical marker genes with two baselines and three existing marker genes selection methods, in the context of cell type mapping.** Given two scRNA-seq datasets with significant batch effect between them, we trained a K-Nearest Neighbor classifier on one dataset (reference), and tested classification accuracy on the other dataset (query). **(a)** Classification accuracies with PBMC control as reference and PBMC stim as query; **(b)** Classification accuracies with PBMC stim as reference and PBMC control as query.

In multiple cell types, the hierarchical marker genes achieved much higher prediction accuracy compared to one-vs-all marker genes.

We also reversed the cell type mapping prediction, treating the PBMC stim dataset as reference and the PBMC control dataset as query. From the cell type prediction accuracies shown in Fig 6b, we again observed that hierarchical marker genes achieved higher prediction accuracy compared to one-vs-all marker genes, in majority of cell types and in average. The average prediction accuracy for one-vs-all marker genes from Seurat and the proposed hierarchical marker genes were 75% and 91%, respectively. This comparison showed that the hierarchical marker genes led to improved cell type mapping accuracy across datasets.

## Discussion

In this study, we proposed a hierarchical marker gene selection approach for interpreting cell clusters derived from scRNA-seq data. Given a clustered scRNA-seq dataset, our approach aimed to construct of a hierarchy of the cell clusters, so that marker genes could be defined for each split in the hierarchy. With a well-constructed hierarchy, our approach was able to identify marker genes for not only individual cell clusters, but also intermediate nodes that represented lineages consisted of closely related cell clusters. In addition, the identified marker genes tended to be more specific to the corresponding cluster or lineage, which was manifested by reduced off-diagonal signals in heatmap visualizations of the identified marker genes.

To evaluate the effectiveness of our proposed hierarchical marker gene selection approach, we compared it with several other gene selection approaches, such as directly using all genes, the highly variable genes, the marker genes selected by one-vs-all approach implemented in Seurat, as well as flat and hierarchical versions of scGeneFit. We applied two strategies for the comparison. The first strategy is to evaluate the selected marker genes within one scRNA-seq dataset. By splitting one scRNA-seq dataset into training and testing sets, a K-Nearest Neighbors classifier was able to quantify how well the selected marker genes could separate cell clusters. We also generated UMAP visualizations that provided a qualitative comparison of separations among cell clusters. The second strategy is to evaluate the selected marker genes across two scRNA-seq datasets that harbored batch effect. Training and testing across different scRNA-seq datasets allowed quantitative evaluation of the robustness of the identify marker genes with respect to batch effects. In both strategies, we demonstrated the benefit of hierarchical marker genes over the one-vs-all flat approach for marker gene selection.

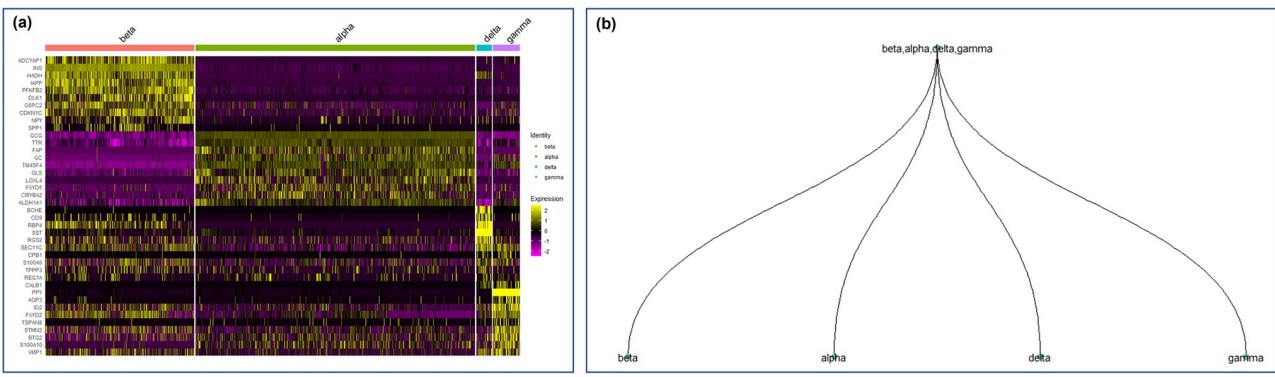

**Fig 7. Marker gene heatmap generated by the one-vs-all FindMarker approach in Seurat.**

To further evaluate our proposed hierarchical marker gene selection approach, we compared the marker genes selected by our approach against those selected by the one-vs-all method. We observed that our method identified a substantial number of genes not selected by the one-vs-all method, particularly within the lineages closer to the bottom of the hierarchy. This indicates that our algorithm effectively identifies new marker genes that can help distinguish closely related cell subtypes. Two specific examples illustrate this capability. In the PBMC3k dataset, our hierarchical method selected *RPL21* as a marker gene for Naïve CD4T. *RPL21* was not selected as a marker gene in the one-vs-all method, and also not selected as a marker gene in higher levels of the hierarchy, which means *RPL21* is a marker gene specifically helpful for separating Naïve CD4 T and Memory CD4 T. Data from the Human Protein Atlas corroborates this, showing higher *RPL21* expression in Naïve CD4 T cells compared to other T cells across three datasets [17]. Similarly, our hierarchical method selected *S100A11* as a marker gene for Memory CD4T. *S100A11* was not selected as a marker gene in the one-vs-all method, not selected as a marker gene in higher levels of the hierarchy. According to CD4+ T lymphocyte reclustering and subcluster analysis in [18], *S100A11* is exclusively expressed by for memory CD4+ T cells, highlighting its role as a specific marker for this subtype.

The advantage of the hierarchical marker genes over the one-vs-all marker genes is dependent on the heterogeneity of the data. When analyzing datasets containing distinct cell clusters without any hierarchical structure, the proposed hierarchical marker gene approach would degenerate to the one-vs-all approach. To provide a concrete example of this scenario, we applied the proposed hierarchical method on pancreas dataset (Xin et al [15]), which included 4 distinct cell types (i.e., alpha cells, beta cells, gamma cells and delta cells). Fig 7 shows the heatmap visualization of marker genes identified by the one-vs-all find marker approach in Seurat, where each vertical section corresponds to one cluster labeled by its corresponding annotated cell type name. Although some off-diagonal expression can be observed in the heatmap, merging the cell clusters did not lead to reduced off-diagonal signal. The proposed hierarchical marker gene algorithm stopped at its first iteration before merging any cell clusters, and the resulting hierarchy was identical to the flat structure of the one-vs-all marker genes approach. Therefore, when there is no hierarchical structure among cell clusters, the proposed hierarchical marker gene approach is the same as the one-vs-all find marker approach. Although the Xin dataset shows one instance where hierarchical and one-vs-all marker genes are equivalent, we believe the proposed hierarchical marker gene approach is advantageous in many biological applications and contexts, because cellular heterogeneity in most biological contexts is hierarchically organized into lineages, cell types and subtypes.

## Methods

### Data preprocessing

We performed data preprocessing using the standard Seurat preprocessing pipeline. We first conducted quality control and removed cells with low gene detection or high mitochondrial gene expression. We then applied library size normalization and log transformation to the data. Following this, we utilized data scaling to eliminate the expression level differences specific to each cell.

### Scoring function

In this study, we introduce a hierarchical marker gene selection method to identify better marker genes for distinguishing cell clusters compared to the popular one-vs-all methods. We proposed a scoring function to determine the hierarchical structure among cell clusters, by evaluating the heatmap visualization of their marker genes. The scoring function aims to encourage new definition of cell cluster groups, such that the one-vs-all selected marker genes based on these cell cluster groups are highly specific to the corresponding cell cluster groups and lowly expressed in other cell cluster groups. Specifically, given a collection of cells and definition of cell clusters for these cells, we first apply the one-vs-all FindMarkers approach in Seurat, and visualize the selected marker genes as a heatmap. We then compute the average the values of diagonal blocks of the heatmap, and sum up all those averages to obtain $V_{diagonal}$. For the off-diagonal blocks, we average the expression values of the marker genes corresponding to a specific cell cluster (or group), and then choose the maximum average value across all cell clusters (or groups). We sum up all the maximum off-diagonal average values across all cell clusters (or groups) to obtain $V_{off-diagonal}$. Finally, we define our scoring function as $s = (V_{diagonal} - k * V_{off-diagonal})/celltype\_number$, where k is a pre-defined parameter that determines the weights of the two terms, in our application, we set k to be 5. Overall, this scoring function measures the expression signals in the diagonal blocks minus large expression signals in the off-diagonal blocks. This is essentially a quantification of the amount of undesirable off-diagonal expression signals. Each split in our cell cluster hierarchy is constructed by iteratively grouping cell clusters to minimize this scoring function, and hence minimizing the undesirable off-diagonal expression signals and making the gene markers highly specific to each cell group.

### Highly variable genes selection

For the purpose of quantitative evaluation of marker gene selection approaches, we decided to examine the highly variable genes, which serves as a baseline gene selection approach for the comparisons. We applied the *scanpy.pp.highly_variable_genes*() function in Scanpy on the preporcessed data to select highly variable genes (HVGs). All the parameters were kept as default settings within the function. In detail, this function calculated the mean and a dispersion measure (variance/mean) for each gene across all single cells, placed genes into 20 bins based on their average expression, performed z-normalization for the dispersions within each bin, and applied a threshold to the z-scores to identify highly variable genes. We found 1406 HVG in PBMC3k dataset, 1398 HVGs in PBMC control dataset, and 1459 HVGs in PBMC stim dataset. For the cell mapping experiment, we kept the 824 overlapping HVGs between PBMC control and PBMC stim.

### Assembled heatmap generation

For each split in our constructed cell cluster hierarchy, marker genes for the branches can be identified and visualized using a heatmap. We concatenated and assembled the marker gene

heatmaps for all splits in the cell cluster hierarchy into one heatmap matrix, where each horizontal section corresponds to an individual split. Other than the first split at the root of the hierarchy, each split only includes a subset of the cell clusters. So, for a particular horizontal section corresponding to one split, expression data for cell clusters not considered in the split are set to 0. The assembled heatmap summarizes all splits of the cell cluster hierarchy, and the data matrix behind this assembled heatmap is what we used to generate the UMAP visualization and classification accuracy in our comparison and evaluation analysis.

## Author Contributions

**Conceptualization:** Yutong Sun, Peng Qiu.

**Data curation:** Yutong Sun.

**Formal analysis:** Yutong Sun, Peng Qiu.

**Funding acquisition:** Peng Qiu.

**Investigation:** Yutong Sun.

**Methodology:** Yutong Sun, Peng Qiu.

**Software:** Yutong Sun, Peng Qiu.

**Supervision:** Peng Qiu.

**Visualization:** Yutong Sun, Peng Qiu.

**Writing – original draft:** Yutong Sun, Peng Qiu.

**Writing – review & editing:** Yutong Sun, Peng Qiu.

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
