## [Decision Letter · Decision Letter 0]

13 Feb 2024

Dear Dr. Qiu,

Thank you very much for submitting your manuscript "Hierarchical Marker Genes Selection in scRNA-seq" for consideration at PLOS Computational Biology.

As with all papers reviewed by the journal, your manuscript was reviewed by members of the editorial board and by several independent reviewers. In light of the reviews (below this email), we would like to invite the resubmission of a significantly-revised version that takes into account the reviewers' comments.

We cannot make any decision about publication until we have seen the revised manuscript and your response to the reviewers' comments. Your revised manuscript is also likely to be sent to reviewers for further evaluation.

Sincerely,

Shihua Zhang

Academic Editor

PLOS Computational Biology

Ilya Ioshikhes

Section Editor

PLOS Computational Biology

Reviewer's Responses to Questions

**Comments to the Authors:**

Reviewer #1: The paper describes a marker selection method that is very relevant to new computational methods in single-cell analysis. Single-cell analysis is moving towards higher granularity of cell types as seen with advances in clonality, sub-clustering, and multi-omics integration. Clonality and genomics data lends itself to creation of tree-like classifications schemes. The method suggested here could be important to define high-resolution differences in clusters, because at a high resolution, the markers that define cell states will differ from markers found at a lower resolution. In addition to suggesting highly specific markers for each cluster in a single cell dataset, the algorithm builds a tree based on these markers, which could be interesting to compare to trees based on clonality, somatic mutations, or other methods for calculating transcriptional similarity.

The paper itself is written very clearly and the scope is well defined. The abstract and summary is concise and accurate, and the idea behind the paper is broadly useful to single-cell genomics.

The method is findable on GitHub and integrated seamlessly with a commonly used framework, Seurat, making it accessible to a broader audience. It is run mostly in R, making it useful for some single-cell analysts, but there are some issues with the code and its scalability. Because of its relevance to recent technological advances and its clarity, it will be useful to the field for discussion and code. I recommend publishing the paper after code revisions.

There are two major issues with the paper.

1. The method of determining the tree based on Wilcoxon tests between all possible combinations of clusters is computationally demanding and slow. In its current state, it is inappropriate for larger datasets. In the code, a Wilcoxon test is run for >1000 cluster comparisons (because many 1 vs. all tests are run) in a dataset with only 9 clusters, making it possible to run for the test case but unscalable to larger numbers of clusters. Theoretically, testing every possible combination of clusters helps build the best possible tree based on markers by using a statistical test for scoring where breaking points should be, providing a highly accurate distance metric between clusters. However, a very similar tree could be found using more computationally frugal methods, for example by calculating distances between centroids of clusters in some transcriptomic space such as PCA space or in the full gene space. There have been methods suggested to calculate these hierarchies that were not mentioned in the paper, namely TooManyCells There is another package that calculates and visualizes a tree for omics data, TooManyCells https://github.com/GregorySchwartz/too-many-cells which creates a tree using “an efficient matrix-free divisive hierarchical spectral clustering”.   I would suggest that these Wilcoxon tests and scores be calculated only once the tree (or an assembly of trees using different methods or from a bootstrap test) is decided, which would reduce the number of Wilcoxon tests required to only N-1 tests, where N is the number of clusters, or close to that number if an assembly of trees are compared to each other. The heatmap and markers could then be decided using the method described here. The method appears to be very useful for scoring markers across a tree of cell-types. It seems important to include the method for building the tree based on Wilcoxon tests as it is currently written, because it is a relevant method for cell type hierarchy calculation and applicable to the reasoning behind the method. But because of the scalability issue, it is important to make the method interoperable with cell-type/cluster trees other than the one calculated directly by the method.   As a starting point I recommend use of the package “data.tree” or “pvclust” (or hclust) to operate on a tree object https://ggraph.data-imaginist.com/. For visualization, I have written some methods for extending the Seurat object to include a tree ggraph object - https://github.com/jo-m-lab/ARBOL. Recent papers on clonality methods use the ggraph package for visualization, so it could be very helpful to somehow integrate this method of marker annotation with ggraph. The TooManyCells mentioned earlier also provides extensive methods for visualizing the tree.

2. The second major issue is that the code provided to run the method is largely undocumented. There is a set of instructions on the GitHub page describing which order to run the code so users can replicate the paper, and I was able to do so, but it is confusing to read the code to check each piece. It also wasn’t clear how the user should input the hierarchy calculated in R into the python notebook. I was able to run the scripts using the instructions on the GitHub page, but it is important for future users that the code is annotated with comments, so that each function can be understood in case of errors and so that it can reach a broader audience.   

3. Lastly, there is one minor issue:

the code requires an older version of Seurat (4.3.0) and does not run with the newest Seurat 5.0. Seurat v5 has caused similar problems for many extension packages, so while I don’t think it’s necessary to update to fit the new package, this should be kept in mind when thinking about accessibility of the method.  

Previous methods for calculating markers highly unique to clusters only use heuristics or basic differential expression to decide how unique the markers are. The scoring method here explicitly calculates markers in a hierarchy of cell types and provides a new scoring method, which could be useful for existing cell-type hierarchy methods. With better documentation, scalability, and interoperability with existing tree objects, the article would be a good read for those in the field.

Reviewer #2: The manuscript is motivated by the non-specified identification of the current one-vs-all strategy of the cell type marker detection on single cell dataset. Given existing cell type annotation, the presented algorithm adopts a hierarchical design by combining the one-vs-all strategy (FinaMarkers in Seurat) with the heatmap of resulting gene markers. The authors defined a scoring function to maximize the sum of the diagonal value and minimize the off-diagonal values in the heatmap. Finally, the author compares the method with traditional Seurat FindMarkers, the scGeneFit (both the flat and hierarchical version).

My major concern is that it is less practical to begin with identified cell types when analyzing a single cell dataset. A more common scenario in this case would start from a set of cells to be annotated, then use clustering to get potential cell types, and ultimately use the cluster markers and existing domain knowledge to annotate the cluster with corresponding cell types. I wonder if this hierarchical marker gene selection method would guide the cell type identification process by enhancing the clustering quality. By factoring the clustering step with the better identified cell markers can really help the researchers with the annotation task.

By looking at Figure 4, the baseline methods with Seurat perform relatively well. I wonder if the authors can compare the actual gene list from both baseline and the proposed method and see if they have overlapped genes. If there are genes specific to the proposed method, how would they be helpful to biological discoveries.

I really like the results for automated cell type mapping which demonstrates a generic usage for constructing a universal classifier for existing cell types on a reference dataset. Again, the ability to transferring the classifier from one dataset to another needs to be further investigated.

Minor comments:

It would be helpful if the authors can draw a workflow diagram illustrating the algorithm.

The heatmap and the hierarchical tree is hard to read due to the font size.

Apply the same color for all heatmaps in the same figure?

It is a little counter-intuitive that the hierarchical version of the scGeneFit always performs the worst among all algorithms. I wonder if the authors can elaborate on this observation.

Can the authors comment on how to obtain the hyperparameter of k = 5.

What is the threshold of z-score in finding the HVG?

Reviewer #3: The manuscript entitled "Hierarchical Marker Genes Selection in scRNA-seq Analysis" presents a computational approach to identification of signature/marker genes for cell populations in scRNA-seq data by simultaneously modeling the cell type hierarchy and their marker genes. With a goal to minimize the off-diagonal expressions, the proposed approach was compared to baseline methods including scGeneFit and one-vs-all seurat method FindMarkers, showing that the proposed approach can improve the accuracy of classifying cell types and mapping cell populations between datasets, using three blood (PBMC) cell datasets.

One advantage of the proposed approach is the assessment of the classification accuracy was done in a cell type-specific way. It clearly shows the differences across multiple methods on different datasets and cell types. The results look promising. There are still a few major issues that need to be clarified to convince the readers about the utility and robustness of the proposed approach, as listed in below:

1) The KNN classifier was trained in 70% of the cells. Were the marker genes identified using the same training set? The testing set needs to remain untouched by the competing marker gene identification methods.

2) There are many classifiers including random forest and SVM. Is there a reason that KNN was selected? In Abdelaal 2019 (Genome Biology), SVM was recommended as the classifier to compare different methods.

3) Besides classification accuracy, ROC curves or precision/recall curves can help show the tradeoff between sensitivity and specificity and compare across competing methods.

4) When there is no cell type hierarchy, the proposed approach seems similar to the one-vs-all approach. This actually shows that the one-vs-all approach is not bad at all. I would suggest to rephrase the sentence that the one-vs-all approach is a special case of the proposed approach. In theory, the one-vs-all approach can be applied to every pair/layer of the cell type hierarchy, if available, to identify the layer-specific marker genes, which is a more fair comparison because in the current setting the cell type hierarchy is not utilized by the one-vs-all approach. The two approaches are just different strategies.

5) The message from Figure 5 UMAP visualization is unclear, especially when compared with using all genes. Consider using a quantitative approach.

6) Describe the data analysis process and (justify) parameters selection used in the assessment/comparison of the competing methods. For example, in the proposed scoring function, is there a way to refine or decide the value for the K (weight) before running the model on the testing set? Is the selection dependent on the abundance of cell populations, number of cells, and number of genes in the dataset? The 3 PBMC datasets seem to be similar in terms of these numbers so it is unclear whether the proposed approach will be robust for different datasets. If the K needs to be selected in an ad hoc way for a new dataset, it will be helpful to include assessment from multiple K values to show the variance.

7) Figure 1 has two sets of a-c. The second set should be d-f. Also the text are too small to see.

8) There are many other marker gene selection methods for scRNA-seq data analysis (SMaSH, COMET, RankCorr, scGeneFit, Seurat, SC3, SCMarker, and scTIM). The related work section needs to be re-organized accordingly to highlight the true innovation of the proposed approach.

**Have the authors made all data and (if applicable) computational code underlying the findings in their manuscript fully available?**

Reviewer #1: Yes

Reviewer #2: Yes

Reviewer #3: Yes

PLOS authors have the option to publish the peer review history of their article (what does this mean?). If published, this will include your full peer review and any attached files.

Reviewer #1: **Yes: **Kyle Kimler

Reviewer #2: No

Reviewer #3: No
---

## [Decision Letter · Decision Letter 1]

5 Oct 2024

Dear Dr. Qiu,

Thank you very much for submitting your manuscript "Hierarchical Marker Genes Selection in scRNA-seq Analysis" for consideration at PLOS Computational Biology. As with all papers reviewed by the journal, your manuscript was reviewed by members of the editorial board and by several independent reviewers. The reviewers appreciated the attention to an important topic. Based on the reviews, we are likely to accept this manuscript for publication, providing that you modify the manuscript according to the review recommendations.

Sincerely,

Ilya Ioshikhes

Section Editor

PLOS Computational Biology

Ilya Ioshikhes

Section Editor

PLOS Computational Biology

Reviewer's Responses to Questions

**Comments to the Authors: **

Reviewer #1: The revised code and article seem widely useful for the computational scRNAseq analysis community. The revised code is much faster and easier to read, thank you! I hope to use this to annotate scRNAseq trees easier.

Reviewer #2: I would like to thank the authors to response to my comments. However, I am still not sure that why the hierarchical version of the scGeneFit performed even worse than the flat version. Also, I wonder if the authors could integrate some of the responses into the manuscript, e.g. the comparison of the markers, the faster implementation, which would help the readers to better understand the merit of the method and have a complete view of the method.

**Have the authors made all data and (if applicable) computational code underlying the findings in their manuscript fully available?**

Reviewer #1: Yes

Reviewer #2: Yes

PLOS authors have the option to publish the peer review history of their article (what does this mean?). If published, this will include your full peer review and any attached files.

Reviewer #1: **Yes: **Kyle Kimler

Reviewer #2: No

Figure Files:

Data Requirements:

Reproducibility:

References:

---

## [Editor Report · Decision Letter 2]

16 Nov 2024

Dear Dr. Qiu,

We are pleased to inform you that your manuscript 'Hierarchical Marker Genes Selection in scRNA-seq Analysis' has been provisionally accepted for publication in PLOS Computational Biology.

Best regards,

Ilya Ioshikhes

Section Editor

PLOS Computational Biology

Ilya Ioshikhes

Section Editor

PLOS Computational Biology

Feilim Mac Gabhann

Editor-in-Chief

PLOS Computational Biology

Jason Papin

Editor-in-Chief

PLOS Computational Biology

---

## [Editor Report · Acceptance letter]

26 Nov 2024

PCOMPBIOL-D-23-01888R2 

Hierarchical Marker Genes Selection in scRNA-seq Analysis

Dear Dr Qiu,

I am pleased to inform you that your manuscript has been formally accepted for publication in PLOS Computational Biology. Your manuscript is now with our production department and you will be notified of the publication date in due course.

With kind regards,

Lilla Horvath
